# Socio-Economic Effect on ICT-Based Persuasive Interventions Towards Energy Efficiency in Tertiary Buildings

**Diego Casado-Mansilla** [1,*], **Apostolos C. Tsolakis** [2], **Cruz E. Borges** [1],
**Oihane Kamara-Esteban** [1], **Stelios Krinidis** [2], **Jose Manuel Avila** [3], **Dimitrios Tzovaras** [2]
**and Diego López-de-Ipiña** [1]

1. DeustoTech—Facultad Ingeniería, Universidad de Deusto, Avda. Universidades, 24, 48007 Bilbao, Spain;
   cruz.borges@deusto.es (C.E.B.); oihane.esteban@deusto.es (O.K.-E.); dipina@deusto.es (D.L.-d.-I.)
2. Information Technologies Institute, Centre for Research and Technology, Hellas, 57001 Thessaloniki, Greece;
   tsolakis@iti.gr (A.C.T.); krinidis@iti.gr (S.K.); dimitrios.tzovaras@iti.gr (D.T.)
3. Wellness TechGroup, Calle Gregor J. Mendel, 6, 41092 Sevilla, Spain; jmavila@wellnesstg.com
*  Correspondence: dcasado@deusto.es

**Abstract:** Occupants of tertiary environments rarely care about their energy consumption. This fact is even more accentuated in cases of buildings of public use. Such unawareness has been identified by many scholars as one of the main untapped opportunities with high energy saving potential in terms of cost-effectiveness. Towards that direction, there have been numerous studies exploring energy-related behaviour and the impact that our daily actions have on energy efficiency, demand response and flexibility of power systems. Nevertheless, there are still certain aspects that remain controversial and unidentified, especially in terms of socio-economic characteristics of the occupants with regards to bespoke tailored motivational and awareness-based campaigns. The presented work introduces a two-step survey, publicly available through Zenodo repository that covers social, economic, behavioural and demographic factors. The survey analysis aims to fully depict the drivers that affect occupant energy-related behaviour at tertiary buildings and the barriers which may hinder green actions. Moreover, the survey reports evidence on respondents' self-assessment of fifteen known principles of persuasion intended to motivate them to behave pro-environmentally. The outcomes from the self-assessment help to shed light on understanding which of the Persuasive Principles may work better to nudge different user profiles towards doing greener actions at workplace. This study was conducted in four EU countries, six different cities and seven buildings, reaching more than three-hundred-and-fifty people. Specifically, a questionnaire was delivered before (PRE) and after (POST) a recommendation-based intervention towards pro-environmental behaviour through Information and Communication Technologies (ICT). The findings from the PRE-pilot stage were used to refine the POST-pilot survey (e.g., we removed some questions that did not add value to one or several research questions or dismissed the assessment of Persuasive Principles (PPs) which were of low value to respondents in the pre-pilot survey). Both surveys validate "Cause and Effect", "Conditioning" and "Self-monitoring" as the top PPs for affecting energy-related behaviour in a workplace context. Among other results, the descriptive and prescriptive analysis reveals the association effects of specific barriers, pro-environmental intentions and confidence in technology on forming new pro-environmental behaviour. The results of this study intend to set the foundations for future interventions based on persuasion through ICT to reduce unnecessary energy consumption. Among all types of tertiary buildings, we emphasise on the validity of the results provided for buildings of public use.

**Keywords:** occupant behaviour; socio-economic profile; survey; energy efficiency; persuasion; intervention; pro-environmental behaviour change; workplace

## 1. Introduction

Energy efficiency and, even more so, energy flexibility approaches related to demand response are inseparably connected with energy-related occupant behaviour. For several years, scientists have tried to tackle the challenge of deconstructing occupant activities and choices in the home and work environment, introducing various behavioural models and adapting previous ones from other scientific fields into the energy sector [1,2]. However, scholars and relevant stakeholders do not always succeed in the study of energy-specific related behaviour. This fact is emphasised in a working environment which seems to be more challenging; working with different user-profiles with different values, beliefs and norms interacting in the same place [3]. Even in recent reviews [4,5], where the high energy-saving potential of occupant behaviour is identified (up to 30% for tertiary buildings), there are still missing aspects that elude the scientific community. These are, but not limited to, systematic frameworks for understanding occupant behaviour that go beyond individual buildings, cultures and geographical boundaries, as well as a more elaborate understanding of persuasion or incentivisation mechanisms for improving more energy efficiency practices. The body of literature reviewed in this article employed questionnaires, interviews or surveys to shed light on those open challenges. These surveys cover a vast area of different parameters but present limitations when trying to accurately pinpoint the factors that affect the behaviour of the occupants towards improving energy performance at working environments.

Energy-related behaviour has been studied for quite a few decades. Over the years, certain theories have been proven to present better results when studying pro-environmental behaviour, with the most well known being Ajzen's planned behaviour theory [6], Hines et al.'s model of responsible environmental behaviour [7] and Stern's value–belief–norm (VBN) theory [8]. As technology improved, behavioural energy approaches have become more sophisticated, integrating the ongoing interaction from multiple drivers/factors [4,5,9]. Yet, despite the efforts on delivering a more integrated scheme, they seem to lack certain aspects, as denoted in the literature. This is especially highlighted by a recent review on energy-related occupant behaviour surveys [10], where in most of the 33 projects that were analysed, essential issues in social science were disregarded, and many other vital aspects of human behaviour were not measured or considered. Some interesting limitations are the lack of in-depth analysis and understanding of (i) the effect that different cultures, countries and climates introduce to occupant behaviour; (ii) the users' actions for restoring comfort conditions; and (iii) the effect that group/collective behaviour may have to energy-related aspects.

If modelling and understanding the user behaviour within buildings is crucial to detect energy leaks and the barriers that prevent them from behaving more energy-efficiently, the same relevance has to be provided to campaigns and interventions to motivate and engage users into green practices. Towards that direction, a new concept in behavioural science has been introduced: the nudge theory (or nudging), which eventually aims to influence the motives, incentives and decision-making of groups and individuals [11]. Investing in this concept, numerous variations have been presented either as tools for deeply understanding factors/drivers that lead end-users to act in a certain way or as a means to persuade and change the overall behaviour. Furthermore, many persuasion mechanisms have been proposed to influence energy-related behaviour towards more energy-efficient actions. Persuasion for sustainability has its roots in the application of Fogg's framework [12] for "computers as persuaders" to the topic of pro-environmental sustainability. Specifically, in interventions based on Information and Communication Technologies (ICT), the use of Fogg's or Cialdini's [13] theories to provide nudges adopted the term "eco-feedback" [14], which, in short, informs users about their actions and make them reflect over resource waste through ambient feedback. Whereas these are widely

provided on health and energy efficiency intervention, persuasion and nudging are not exempt from controversy. There are recent works which lower the impact they have [15,16] or voices that raised their concern about the feasibility of persuasion to maintain the target behaviour in the long-term [17,18]. In short, going beyond understanding energy-related behaviour, when trying to persuade end-users to alter their energy-expensive behaviour, the proper persuasion tools that have the desired effect to occupants—even more in cases of tertiary buildings—still intrigue the scientific community.

In order to link the two presented challenges, this work presents a survey that takes into account both socio-economic factors and energy-related aspects to better understand the human drivers that could be integrated on a new ICT-based persuasion-based engine for improving occupant behaviour at an individual's working environment. The survey responses help identify which profiles and principles are more likely to instil behavioural change towards a more energy-efficient practice at work by building a socio-economic profile for each of the occupants of the workplace. One of the main particularities of the survey is that it was conducted in seven different buildings across six different cities in four EU countries and reaching more than three-hundred-and-fifty people (350). Therefore, the analysis of the dependencies between factors and principles takes a broader perspective considering the application of the questionnaire to people with a very different background and a heterogeneous socio-demographic profile. Further refinement of the survey was also elaborated after the assessment of a behavioural change intervention, highlighting the factors that were identified in the beginning but did not present significant results after the first iteration of responses.

The presented work is structured as follows. Section 2 presents the GreenSoul project and provides the basis of the intervention we carried out, followed by Section 3 that explains the design of the survey performed. Section 4 delivers the statistical results, and Section 5 provided a thorough discussion on them along with the limitations of the study. Finally, Section 6 concludes the manuscript containing a final outlook of research.

## 2. The GreenSoul Project

In an effort to achieve higher energy efficiency in public buildings, the GreenSoul (GS) project [19], an H2020 research and innovation action, designed, implemented and tested a system for saving energy in tertiary buildings allowing to produce energy savings up to 25% in the best scenario [20]. GS is based on monitoring devices of personal (e.g., laptops or monitors) and shared equipment (e.g., lighting, HVAC or appliances) and mechanisms to give personalised usage recommendations and nudges, taking into consideration the characteristics of the users and adapting the feedback to their changes (the feedback is different depending if the user should form, enhance or maintain an eco-behaviour). One of the hypotheses of the project is that each user-profile has an optimal way to receive feedback on their energy consumption and good practices to increase energy performance. Therefore, profiles are identified through online questionnaires and the system makes use of this information to create a tailored socio-economic model of the user to incentive him/her to save energy through innovative ICT elements. By investing in end-user engagement, rather than sophisticated automation or Building Management Systems (BMS), the cost is reduced and the results shall be prolonged. To encourage users to reduce energy consumption, an additional effort has been included to persuade them to be energy efficient (the impact of daily habits on the energy consumed at work is mainly reflected in the electricity bill, which never reaches the employees). To this end (a) the behaviour of people related to energy efficiency at work is modelled; (b) a socio-economic characterisation and segmentation is carried out using statistical analyses; (c) energy monitoring hardware was deployed in each of the six pilots; and finally, (d) the effectiveness of different ICT-based persuasion strategies provided through different interactive channels was tested in each of the identified population segments.

### 2.1. The GreenSoul Intervention

To test the effectiveness of the overall GreenSoul system, a Randomised Control Trial (RCT) was carried out in seven pilot buildings across Europe involving more than three-hundred-and-fifty people

(350). Four different treatments combining three different persuasion principles (i.e., cause–effect, self-monitoring and conditioning) through ICT were deployed. These treatments were delivered using different feedback channels: a custom-based interactive coaster that provided visual information about energy consumption (self-monitoring), a gamified mobile app with some automation features (conditioning), a series of analogue signage in the form of post-its and posters with "green messages" (cause–effect), which can be considered as the control-treatment, and all of the three previous treatments altogether. Figure 1 shows a picture of them.

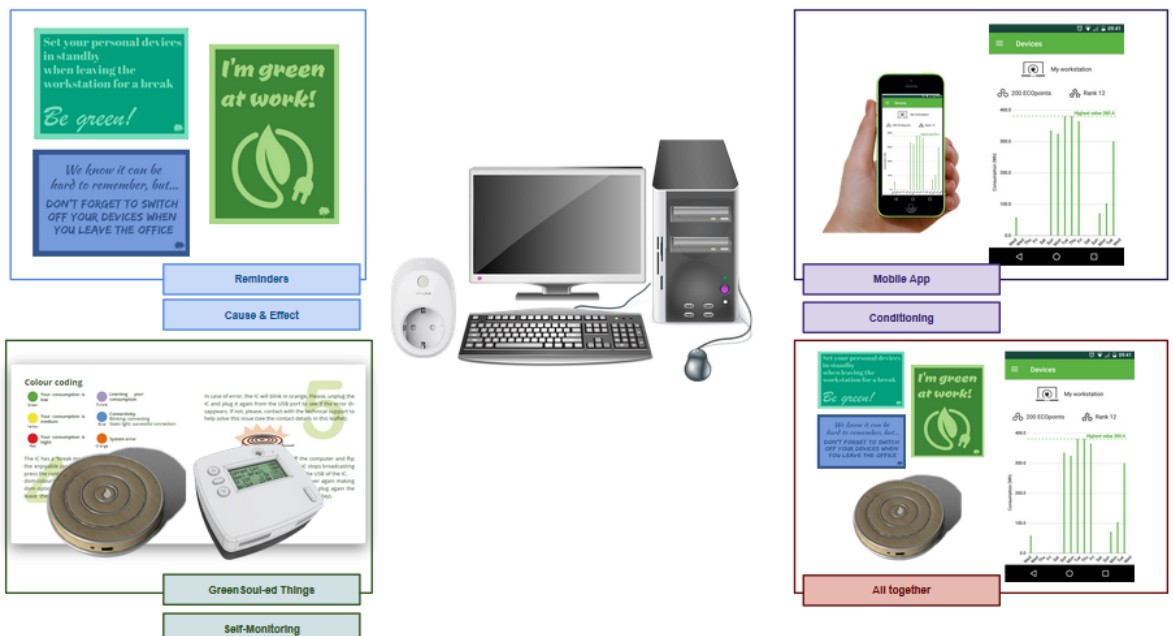

**Figure 1.** The GreenSoul persuasion treatments.

We prepared the intervention on the basis of the following hypothesis; individualised persuasion messages based on personal energy consumption produce a high engagement, but the potential impact is low compared to the energy consumption of the whole building. However, if an intervention started addressing directly the collective, its potential impact on energy-savings is higher, but end-users might not be motivated to pursue green-actions (see Figure 2). To overcome this problem, the RCT was divided into two phases: individual and collective. During the individual phase, the primary objective was to foster the awareness and motivation of the participants in energy efficiency practices. Therefore, we only provided individual information regarding their performance with devices and appliances under their own control. In the second phase (with all participants already engaged), we gave persuasive cues about how to reduce the energy consumption of electricity-powered devices not directly attached to the individual but more related to equipment of shared use (e.g., lighting, HVAC or common appliances). In workplaces, there is usually no one in charge of them and the diffusion of responsibility and social loafing appear as the primary factors for energy waste [21]. Indeed, according to Whittle and Jones [22], "Both theories indicate that in group situations people are generally less likely to take action than when alone because, for instance, they do not assume personal responsibility for taking action (anticipating that someone else will do so) or because they believe that others will not put in the effort, so do not put in (as much) effort themselves."

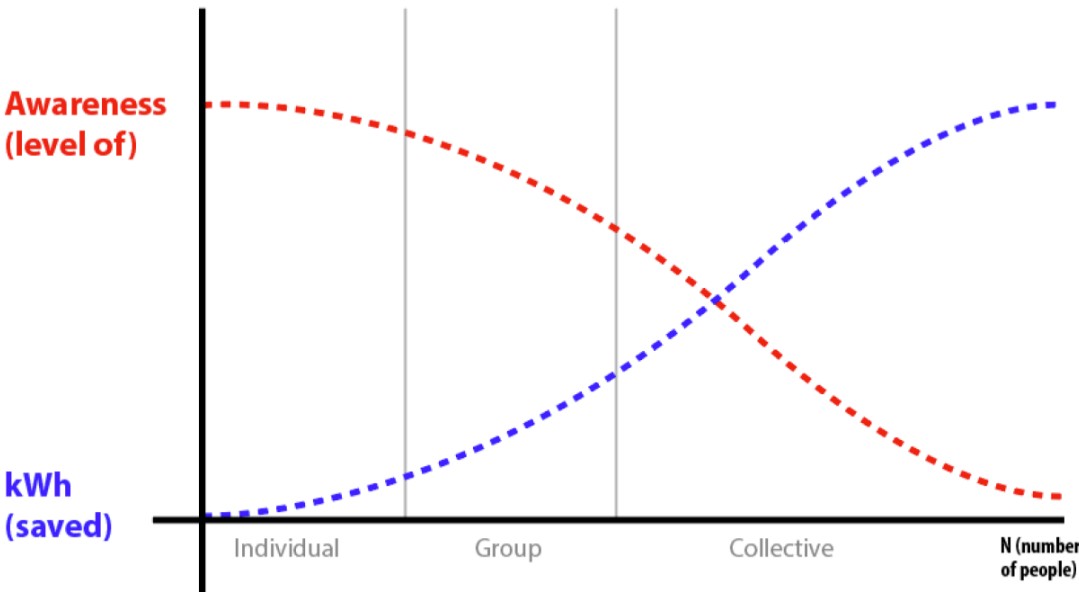

**Figure 2.** The GreenSoul phases: Individual where motivation is high but the effect on energy efficiency and reduction is low. Collective, where the motivation of users to embark on campaigns might be low, but the potential of the energy savings is high.

*2.2. Evaluation Procedure*

We triangulated the overall GreenSoul solution from three different qualitative and quantitative sources: (1) PRE-POST validated surveys to assess energy awareness, motivations to change the behaviour and main obstacles that hinder the adoption of energy practices in the workplace; (2) the energy consumption per user, per treatment and per building along the whole study; (3) focus groups throughout the whole experimental phases to understand user motivations at each time, interventions pitfalls and other relevant matters. In this article we put the focus on the questionnaires to provide a thorough analysis of the user profiles and their specific motivations and barriers to behave pro-environmentally at the workplace.

## 3. Survey Design and Delivery

The survey presented in this manuscript was developed in two stages: the PRE and POST pilot interventions. Starting with a state-of-the-art analysis, validated instruments and acquired knowledge from previous experiences, the initial survey was constructed, before applying any interventions to the buildings, consisting of 36 groups of questions divided into four sections [23]. The final outcome provides a complete demographic and socio-economic outline of the respondents including also a section for their self-rate preferences in terms of what persuasion strategies they would like to be applied in an energy-related intervention in the workplace. The PRE-pilot full four sections are briefly explained in the following sections. A more elaborated description is provided in [24].

*3.1. Social, Economic and Demographic Traits*

Questions related to the personal information of the users are essential towards being able to identify general groups and profiles that may be more keen to change based on the suggestions provided. We subdivide such information into those factors related to the workplace and those which are independent.

1. *Non-Dependent on Work*: Common questions that cover demographic (static) aspects such as age, gender, education, country and city. Additionally, this group includes questions related also to dynamic aspects such as confidence in technology, susceptibility to persuasion as well as attitudinal and intentional profiles.

2. *Dependent on Work*: Questions directly influenced by the organisation where the user is employed, consisting once again of static information such as the type of employment, the position, work culture, etc. and dynamic energy-related factors such as the main barriers to be energy efficient, the willingness to join an initiative to reduce energy consumption, and more.

3. *Energy-related actions at work*: An ancillary profile was constructed through questions that cover energy-related social habits such as the use of the HVAC, the set-point established in summer and winter, the use of natural and artificial lighting, printing habits or the power mode set in users' equipment.

4. *Ranking of Persuasion Strategies*: Users were asked to self-rate (one to five likert scale) 21 persuasive strategies related to enhancing energy efficiency practices in shared spaces at working environments. The strategies were designed based on fifteen persuasion principles developed by experts [12,13,25]. Table 1 presents the Persuasion Principles, Table 2 introduces the Persuasion Strategies (numbered from v2 to v22 as this is the coding they have on Zenodo's dataset), and in Table 3 these two are mapped together.

In order to evaluate and validate the interventions of the GreenSoul project, a second survey was constructed after the completion of the pilot execution period. Given the findings originating from the analysis of the first survey, a lot of socio-economic questions were omitted in the second version (e.g., country, having children or not, current employment status, are you satisfied with your thermal comfort at workplace, etc.), as they were not found to have significant influence to any of the Persuasive Principles under study. Besides, we removed the five least ranked principles of persuasion by initial respondents to only provide again those that might have higher impact influencing people from their point of view. Finally, we added ten questions aiming to identify the perspective of end-users regarding the project and its effect.

The two surveys designed for the PRE-pilot and POST-pilot assessments along with their raw results can be found on Zenodo in [23] and [26], respectively.

**Table 1.** Explanation and Rationale of 15 Persuasion Principles.

| # | Persuasion Principle | Description |
|---|---|---|
| P1 | Authority | People will tend to obey authority figures, even if they are asked to perform objectionable acts. People want to follow the lead of real experts. |
| P2 | Cause and effect | This principle can effectively persuade people to change their attitudes or behaviour by enabling them to observe a direct link between cause and effects of daily actions. |
| P3 | Conditioning | A behavioural process whereby a response becomes more frequent or more predictable in a given environment as a result of reinforcement, with reinforcement typically being a stimulus or reward for a desired response. Computerised systems use principles of operant conditioning to change behaviours. To use operant conditioning is basically to provide reinforcement (positive or negative) to the user. |
| P4 | Cooperation Liking | A system can motivate users to adopt a target attitude or behaviour by leveraging human beings' natural drive to cooperate. Further, People prefer to say "yes" to those they know and like. People are also more likely to favour those who are physically attractive, similar to themselves, or who give them compliments. Even something as "random" as having the same name as your prospects can increase your chances of making a sale (e.g., Coke). |
| P5 | Tailoring Personalisation | A system that offers personalised content or services has a greater capability for persuasion. Further, information provided by the system will be more persuasive if it is tailored to the potential needs, interests, personality, usage context, or other factors relevant to a user group. It provides information that is specific to the individual to better enable a certain behaviour. |
| P6 | Physical attractiveness | It plays an essential role in making the application easy to understand and more likeable. |
| P7 | Praise | By offering praise, a system can make users more open to persuasion. |
| P8 | Real-world feel verifiability | A system that highlights people or organisation behind its content or services will have more credibility. System should provide information of the organisation and/or actual people behind its content and services. Moreover, credibility perceptions will be enhanced if a system makes it easy to verify the accuracy of site content via outside sources. |
| P9 | Reciprocity | Reciprocation recognises that people feel indebted to those who do something for them or give them a gift. People tend to return a favour. |
| P10 | Reduction | A system that reduces complex behaviour into simple tasks helps users perform the target behaviour, and it may increase the benefit/cost ratio of a behaviour. |
| P11 | Self-monitoring | A system that keeps track of one's own performance or status supports the user in achieving goals. The goal is to allow people to monitor themselves to modify their behaviour to achieve a predetermined objective or outcome. |
| P12 | Similarity | People are more readily persuaded through systems that remind them of themselves in some meaningful way. We are more persuaded by people we think are similar to us (personality, interests, etc). |
| P13 | Social proof | A psychological phenomenon where people assume the actions of others in an attempt to reflect correct behaviour for a given situation. It is the concept that people will conform to the actions of others under the assumption that those actions are reflective of the correct behaviour. |
| P14 | Social Recognition | By offering public recognition for an individual or group, a system can increase the likelihood that a person/group will adopt a target behaviour. |
| P15 | Suggestion | Systems offering fitting suggestions will have greater persuasive powers. People are more likely to engage in an activity when it is closely related to what they are currently doing. |

**Table 2.** Description of the 21 Persuasion Strategies Identified.

| # | Persuasion Strategy |
|---|---|
| v2 | Public (social) recognition of your contribution to energy savings is provided |
| v3 | Receive personal praise (privately) for your contribution to energy savings |
| v4 | The support of the majority of your peers to improve energy efficient behaviour |
| v5 | Receive energy related information in a simple and aesthetically appealing way |
| v6 | Receiving perks such as flexible working hours, skipping certain tasks, etc., as a reward for improving your energy performance |
| v7 | You and your team receive recognition for collectively achieving energy savings |
| v8 | You receive information about the people (e.g., engineers, vendors, etc.) behind the instruments and equipment which allows you to collect energy-related data |
| v9 | You are assisted in setting, meeting and reviewing your own personal energy saving goals |
| v10 | Your (top) managers are also committed to save energy |
| v11 | You can monitor & track your own energy performance in real-time |
| v12 | The overall energy saving goals are broken down into smaller easily achievable |
| v13 | The feasibility of the proposed energy savings has been verified in other buildings similar to your workplace |
| v14 | Energy related information is tailored to you and you are able to self-configure some parameters (e.g., data provided, frequency, etc.) according to your preferences |
| v15 | Information on the actual effect that your (potential) actions may have upon the energy consumption |
| v16 | Comparative assessment of your actual energy performance compared to benchmarks/ good practices |
| v17 | Comparative assessment of your energy saving performance with the respective performance of your peers (e.g., colleagues, other visitors, etc.) |
| v18 | Historical comparison of your energy performance and/or consumption |
| v19 | Tips or suggestions on the energy saving practice of the day/week |
| v20 | Progress, tips and lessons learned on specific energy saving actions performed by other users that are similar to me |
| v21 | Advice and quotes from energy experts (including external energy consultants, energy researchers, energy agencies, etc.) |
| v22 | Links to data about how energy consumption is monitored and (potential) energy savings assessed |

**Table 3.** Mapping of persuasive strategies to the persuasion principles.

| Persuasion Principle | Persuasion Strategy |
|---|---|
| Authority (P1) | v10, v21 |
| Cause and effect (P2) | v15 |
| Conditioning (P3) | v6 |
| Cooperation & Liking (P4) | v4 |
| Tailoring & Personalization (P5) | v9, v14 |
| Physical attractiveness (P6) | v5 |
| Praise (P7) | v3 |
| Verifiability & Real-world feel (P8) | v8, v13, v22 |
| Reciprocity (P9) | v7 |
| Reduction (P10) | v12 |
| Self-monitoring (P11) | v11, v16, v18 |
| Similarity (P12) | v20 |
| Social proof (P13) | v20, v10 |
| Social Recognition (P14) | v2, v7 |
| Suggestion (P15) | v19 |

*3.2. Survey Setup*

An online survey was delivered in seven official tertiary pilot buildings in the following cities; Bilbao—Spain, Cambridge—UK, Sussex—UK, Pilea-Hortiatis—Greece, Seville—Spain, Thessaloniki— Greece and WEIZ—Austria. The survey was delivered in two stages, prior the intervention of the GreenSoul framework (Pre-pilot—between May and July 2017) and after it (Post-pilot November 2019) (A description of the pilots is provided in the Appendix A of this manuscript). For the GreenSoul pilot execution, only five out of the seven buildings were used, hence allowing the existence of control groups (in our case ECOLUTION (Sussex) and CERTH (Thessaloniki)).

In total, for the Pre-Pilot survey three-hundred-and-twenty-three (323) responses to the questionnaires were collected. After conducting a data cleaning process (i.e., removing uncompleted questionnaires and outliers) three-hundred-and-three (303) samples remained to be analysed. Of these responses, eighty-four (84) correspond to people working in the Greek sites (PILEA & CERTH), eighty-three (83) were those working in Spain sites (DEUSTO & SEVILLE), eighteen (18) were those working in Austria (WEIZ) and, most of the answers, one-hundred-eighteen (118) respondents working in the UK (ALLIA & ECOLUTION). Accordingly, for the Post-Pilot survey after two years of intervention, one hundred and five (105) responses were retrieved from the pilots. Of these responses, thirty-seven (37) correspond to people working in the Greek sites, forty-three (43) were those working in Spanish sites, twelve (12) were those working in Austria and thirteen (13) respondents working in the UK. Table 4 summaries the number of completed questionnaires collected per pilot site per survey trial period. The surveys were deployed online using Google Forms, and they were distributed through email to all sites in their national language. The participation was voluntary and a pilot responsible provided reminders during the survey period to ensure an adequate participation rate per site.

**Table 4.** Number of responses per GreenSoul cities

|            | Bilbao | Cambridge | Sussex | Pilea | Seville | Weiz | Thessaloniki |
|------------|--------|-----------|--------|-------|---------|------|--------------|
| **Pre-Pilot**  | 53 | 58 | 60 | 26 | 30 | 18 | 58 |
| **Post-Pilot** | 28 | 8  | 5  | 19 | 15 | 12 | 18 |

## 4. Results

The analysis of the responses collected has been performed in both a descriptive and a prescriptive manner. In the former, the analysis aims to provide some basic descriptive statistics and conclusions over the data collected. Regarding the socio-economic and demographics, we can conclude which are the prominent profiles among different pilot buildings, whether we find gender or technical divide in the population or if our sample is willing to join energy initiatives at work or they are reluctant. In relation to the persuasive part, we can conclude which are the most and least rated persuasive strategies and the principles behind them. The prescriptive analysis is devoted to identifying socio-economic factors that are potentially important in determining certain persuasion principles to be used or which profiles would be more prone to elicit changes in pro-environmental behaviour according to persuasive strategies behind these principles.

### 4.1. PRE/POST Preprocessing

Before analysing the results from the two surveys, a preprocessing of the received data was necessary. For the first survey, given the size of the overall questionnaire, a dimensionality reduction of the overall system was decided. All the factors related to habits in the workplace (i.e., Ancillary profile) were excluded, as they are tightly dependent on the workplace where employees were at the moment of answering the questionnaire. Therefore, we finally analysed the socio-economic profile with the following variables; Age, Gender, Education, Country, City, Employment, Position, Work culture, Sharing (whether people share the room at work with others or not), Work activity, Profile PST (Pinball, shortcut, thoughtful) [27], Intentions (to behave pro-environmentally), Confidence (in technology), Organisation energy (if employees perceive their organization as green or not), Barriers (to behave pro-environmentally in the workplace), Consensus (difficulty or ease of reaching consensus with peers), Influences (if people tend to influence others or not), Susceptibility (of people to persuasion), Initiative to join (environmental campaigns), desired Frequency to receive feedback and Response provided to green signage.

Regarding the persuasive strategies, per user we found the sub-set of Persuasive Principles that were rated with the highest rank, and we distributed one score among them. Thus, if an employee rated $N$ principles, we assigned $1/N$ points to each principle. After finalising the process, we came up

with a dataset of principles of persuasion with a score in only those which were voted with the top rank. Finally, questions that were not answered were not included in the analysis.

*4.2. Descriptive Analysis: PRE vs. POST*

As the PRE-pilot descriptive analysis has been described thoroughly in [24], the comparative analysis of the two surveys is presented next highlighting the most interesting aspects identified in both timeframes.

In the POST-pilot survey analysis a slight shift is observed towards male (61.9%) over female (37.1%) respondents, whereas the PRE-pilot was more balanced with 48.1% women and 51.9% men. The principal age group remained at the range of 21 to 40 (49.5%), whereas the percentage of participants holding a master's degree has elevated by 10% (POST: 41.0% over PRE: 31.7%), and the respondents that have only finished high school has slightly diminished (POST: 4.8% over PRE: 5.1%). The position has been divided in more groups with 73.33% working as employees (compared to the previous PRE: 88.5%), 8.57% as higher positions (principal researchers, head of unit or boss), and 16.19% as administrative staff. Following the same analysis as the PRE-pilot questionnaire, the profiles identified, according to Lockton's model of the user [27], were in the following order; Thoughtful (POST: 44.21% instead of PRE: 54.3%), followed by Pinball (POST: 15.79% instead of PRE: 11%) and Shortcut (POST: 14.74% instead of PRE: 14.3%). Therefore, we observed that the differences among the two snapshots in PRE and POST do not vary significantly in socio-economic terms. Thus, for this study, we assume that the two samples could be considered as comparable units for evaluation purposes. In the limitations section at the end of the manuscript we elaborate on this issue.

In terms of persuasion principles, even though fifteen principles were originally identified by experts, most of them were not found to be important enough for end-users to receive significant attention from them. As a result, in the POST-pilot survey, only the top ten principles that were highly rated in the PRE-pilot were included. As can be observed in Figure 3, there were quite a few differences when rating these by the end-users prior and after the interventions. However, we can observe that Cause and Effect (P2), Self-monitoring (P11) and Conditioning principle (P3) were top-rated in both surveys.

From the answers received about employees' intentions to behave in favour of the environment using as instrument the validated questionnaire provided by [28], an interesting increase has been observed in the POST-pilot survey in the Action stage (PRE: 47.84% vs. POST: 56.38%), followed by a reduction in the Contemplation (PRE: 48.17% vs. POST: 38.30%). These evaluated stages are alike to those extracted from Transtheoretical Model of behaviour change (TTM) [29], which can be observed in Figure 4.

According to the provided results, it seems that end-users have shifted from hesitation to real determination to act in a more energy efficient manner after the GS intervention was provided. Statistically speaking, we found an important difference in Pre-Contemplation stage between the responses of PRE and POST pilots questionnaires when treating the dataset as a whole (i.e., pooled data). Wilcoxon Signed-Ranks Test indicated that the median pre-test ranks were statistically significantly higher than the median post-test ($W = 1, Z = 4.5399, p < 0.05, r = 0.2225$). Thus, the average mean of people in Pre-Contemplation stage before the treatments was greater than the computed average at the end of the interventions. Thus, the overall employees participating in GreenSoul project seem to have reduced their stage of Pre-contemplation in favour to higher stages, where the willingness to do action in favour of the environment increased. This observation has a medium effect size according to Cohen's criteria [30].

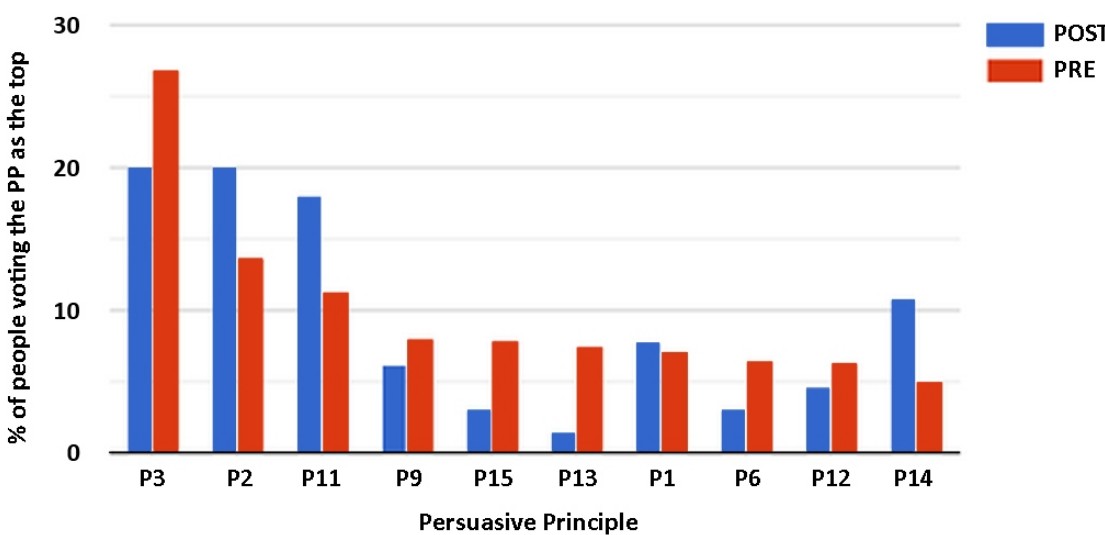

**Figure 3.** PRE vs. POST-pilot: Ranking of the top ten dominant persuasion principles as voted by the end-users as a top principle.

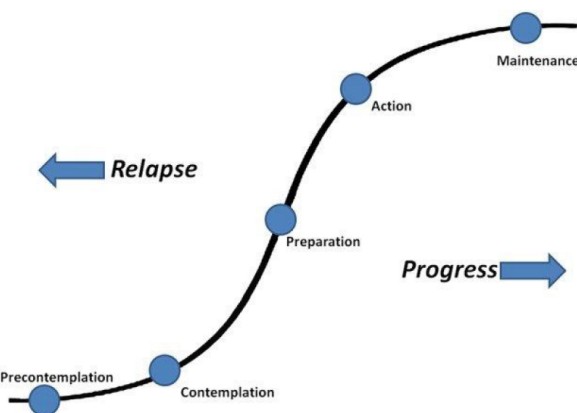

**Figure 4.** The stages of change according to the transtheoretical model to evaluate enhancement or relapse of behaviour change.

Descriptive results so far covered the respondents' opinions using a pooled sample approach. When examining pilot by pilot (recall that a description of the pilots is provided in Appendix A), some interesting outcomes were observed.

### 4.2.1. SEVILLE

A contingency analysis for the barriers using Fisher's test [31] showed that people in this pilot increased their overall intentions in favour of the environment significantly ($p$-value < 0.02). Specifically, we can observe in Figure 5 that the percentage of people in the Contemplation stage (2) was reduced and increased in Action stage (3) in the POST-pilot results. Pre-contemplation (1) in PRE and POST was similar.

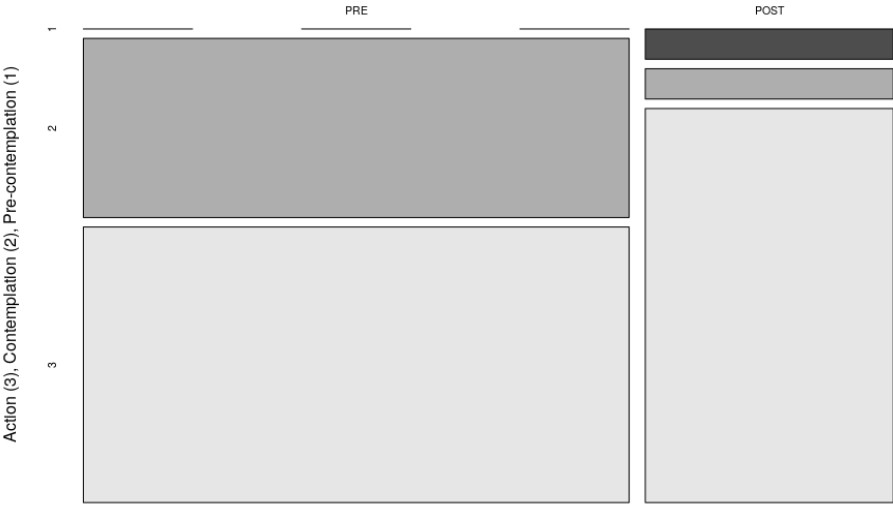

**Figure 5.** Overall pro-environmental Intentions in Seville (PRE vs. POST-pilot). We can observe the reduction of people in contemplation stage and the relevant increase of people reporting to be in Action.

### 4.2.2. DEUSTO

We observed an important difference in Pre-contemplation stage between the responses of PRE and POST pilot questionnaires in this Spanish pilot ($W = 1, Z = 5.1428, p < 0.05, r = 0.5749$). According to the large effect size observed ($r = 0.5749$) with regards to Cohen's criteria [30], the results might suggest that this pilot contributed importantly to the effect observed above in the pre–post analysis when examining the pooled samples approach. Furthermore, the tests in Contemplation and Action stages confirmed that not only people in this pilot scored lower on Pre-contemplation, but they also increased their stages of change towards more conscious eco-behaviour. Contemplation ($W = 1, Z = -2.0434, p < 0.05, r = 0.228$) and Action ($W = 1, Z = -2.4688, p < 0.05, r = 0.5749$).

### 4.2.3. WEIZ

According to the results, the medians of PRE and POST regarding Pre-contemplation stage in Weiz were 0.2 and −1.47, respectively. A Wilcoxon Signed-rank test shows that there is a significant effect of Group WEIZ with the Pre-contemplation stage of the TTM ($W = 1, Z = 1.9598, p < 0.05, r = 0.3464$). That means that the respondents change their Pre-contemplation stage towards higher stages of consciousness. Interestingly, Contemplation also differed statistically ($W = 1, Z = 2.226, p < 0.05, r = 0.3934$), but in the opposite direction. Thus, less people self-assessed their stage of change in Contemplation. As we did not observe a significant change in Action stage nor in the overall intentions, we concluded that the interventions helped WEIZ's employees to be aware of the energy concerns, but no more.

### 4.2.4. ALLIA

Similar observations occurred in Cambridge, UK (ALLIA). The medians of PRE and POST regarding Pre-contemplation stage were −1.25 and −2.75, respectively. A Wilcoxon Signed-rank test shows that there was a significant effect as well ($W = 1, Z = 2.360, p < 0.05, r = 0.2905$). This means that ALLIA's employees equally changed their Pre-contemplation stage towards higher stages of consciousness.

#### 4.2.5. MPH

According to the questionnaires, employees in Pilea-Hortiatis (Greece) reported statistical evidence that the mean of the Contemplation stage was different between PRE ($\mu = 1.074$) and POST ($\mu = -0.55$) ($W = 1, Z = 2.1388, p < 0.05, r = 0.356$). However, as not conclusive data for predecessor or subsequent stages were found, we can not conclude that MPH's employees improved or worsen their willingness to do actions in favour of energy efficiency.

A contingency analysis for the barriers using Fisher's test showed that significantly ($p < 0.01$) none of MPH's participants reported that their hurdles to behave energy efficiently were due to a lack of knowledge after the delivery of the treatments in POST (the evaluated item: "I am not sure about what is a good energy practice so I do little or nothing") nor people were found to be discouraged about peers at work (the evaluated item: "I am discouraged by the attitude of my colleagues and/or of the management, so I do little or nothing"). The following Table 5 provides an overview of these findings. These results on barriers were only reported in Pilea's pilot and not observed in other premises.

**Table 5.** Contingency table of MPH: PRE vs. POST in overall barriers to behave energy efficiently.

|      | Absentmindedness | Lack of Awareness | Peers' Discouragement | Other |
|------|------------------|-------------------|-----------------------|-------|
| PRE  | 6                | 5                 | 5                     | 10    |
| POST | 10               | 0                 | 0                     | 3     |

#### 4.3. Prescriptive Analysis

The prescriptive analysis of the survey results aims at identifying and quantifying interactions between socio-economic data and persuasion strategies or other important behavioural constructs such as Pre-Contemplation, Contemplation, Action, Intentions, Barriers and Confidence in technology. To this aim, it was decided to use contingency tables which are popular in surveys' evaluation [32], also known as cross tabulation, as a means to understand whether the top rankings provided to PPs could be dependent on the variability of each socio-economic factor under study. Next, we provide these associations at two different snapshots separated by one year and a half: PRE and POST.

#### 4.3.1. Results from Pre-Pilot Questionnaires

We evaluated if any of the fifteen Persuasive Principles were dependent on studied socio-economic variables (e.g., Gender: male/female). A Pearson's chi-squared test was applied to evaluate how likely it is that any observed difference between the variables within each factor arose by chance. To be even more rigorous and conservative, as multiple hypotheses were tested, a Bonferroni correction was applied on the significance levels for validating the hypotheses.

The results obtained can be observed in Table 6. The table shows that the only factors which show dependencies on some principles of persuasion were: City, Education, and Initiative_to_join. This latter factor, as well as City, were the ones which presented more significant interactions with different Principles of Persuasion. Specifically: Cause & effect (P2), Praise (P7), Similarity (P12) and Suggestion (P15).

Furthermore, we investigated if there were dependencies among the city of origin and certain important factors, such as Pre-Contemplation, Contemplation, Action, overall Intentions, Barriers and Confidence in technology. This test is relevant since it helps to observe if different employees respond differently to some factors under study depending on their country of origin, the city where they live or the working area where they are employed.

**Table 6.** Significant *p*-values (and the associated power) that ascertain the dependencies between factors and persuasive principles for the PRE survey.

| Persuasion Principles | City | Education | Initiative_to_join |
|---|---|---|---|
| **P2** | $p < 0.0005$ $\beta = 0.9944237$ | - | $p < 0.0005$ $\beta = 0.9339491$ |
| **P7** | - | $p < 0.005$ $\beta = 0.9188609$ | $p < 0.002$ $\beta = 0.9659801$ |
| **P12** | $p < 0.001$ $\beta = 0.9764619$ | - | - |
| **P15** | $p < 0.0005$ $\beta = 0.9996468$ | - | - |

Pre-Contemplation, Contemplation, and Action

A Kruskal–Wallis test between the city and these three factors revealed a significant effect of $(\chi^2(1) = 462.69, p < 0.01)$, $(\chi^2(1) = 285.7, p < 0.01)$ and $(\chi^2(1) = 303.3, p < 0.01)$. In Table 7 the results from a post hoc analysis is provided using Dunn's tests with Bonferroni correction.

**Table 7.** PRE-pilot: Post hoc analysis to identify where the differences are in the effect that we observed after running Kruskal-Wallis test. In Pre-contemplation it seems that Weiz scored differently to other pilots. Specifically, people in Weiz pilot scored higher the Pre-contemplation stage than the rest of the pilots.

| | Pre-Contemplation | Contemplation | Action |
|---|---|---|---|
| **City** | Weiz-Bilbao Weiz-Seville Weiz-Thessaloniki Weiz-Cambridge Weiz-Sussex $(p < 0.002; r = 0.0875)$ | Bilbao - Sussex $(p < 0.025; r = 0.058)$ | - |

Intentions, Barriers and Confidence in Technology

A contingency analysis using G test (The G test statistic is also approximately chi-squared distributed, but for small samples. This approximation is closer than one that chi-squared test uses) showed that Confidence in Technology $(\chi^2(14) = 25.763, p < 0.05)$ and the different barriers to behave pro-environmentally $(\chi^2(21) = 54.117, p < 0.01)$ also depend on the city significantly. In the following figures the reader can observe the distribution of different responses depending of the geographical distribution of pilots. Those images help identifying in a glimpse which of them responded differently. The area of tiles in the figures is proportional to the number of observations within each category (The reader can observe that number four is missing in the Figures (i.e., Figures 6 and 7). This is because we provided the codification of number four for other cities. However, as we did not get any observation from other locations, we show no data for number four).

According to the Pearson standardised residuals measures [32] performed over the contingency data in Figure 6, it indicates that the category Confidence in technology is negatively associated with people in Sussex pilot and No-Confidence positively associated with the employees of this UK building. Moreover, Seville's employees presents a positive association with No-Confidence and Thessaloniki's respondents presents a negative association with No-Confidence. These results means that in Sussex there are few people associated with Confidence in technology and many that are associated with No-confidence. The majority of Seville respondents are associated to No-Confidence, in contrast with Thessaloniki where just a few are associated to No-confidence (Pearson standardised residuals measure how large is the deviation from each cell to the null hypothesis (in this case, independence between row and column's). Please note that results with absolute value greater than two are significant indicative of association).

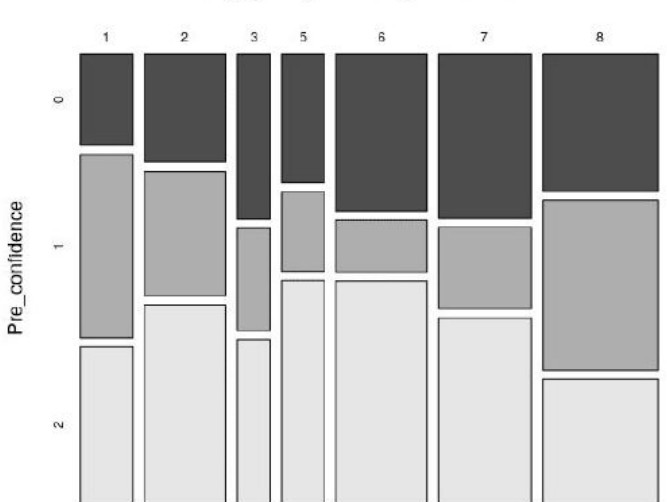

**Figure 6.** PRE-pilot: Comparison of the distribution of responses to confidence in technology among different pilots: neutral (0), No-Confidence (1) and Confidence (2).

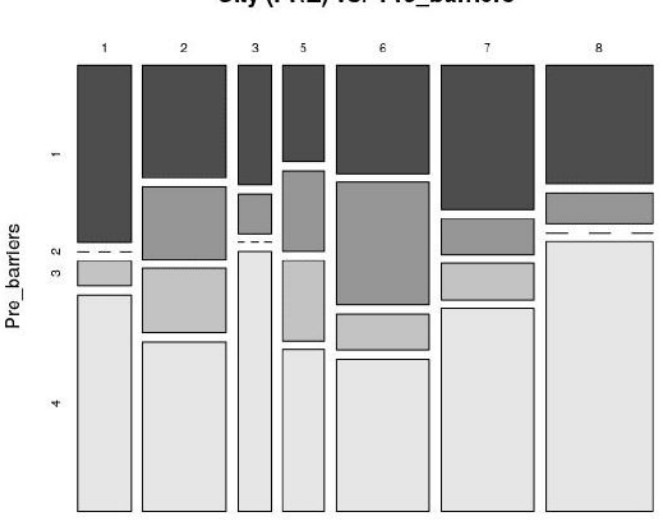

**Figure 7.** PRE-pilot: Comparison of the distribution to barriers encountered in each pilot: Absentmindedness (1), Lack of Awareness (2), Discouragement from Peers (3) and Other Reasons (4). In a glimpse we can observe that there are differences among certain pilots.

According to the Pearson standardised residuals measures performed over the contingency data in Figure 7, we observed a significant negative association of factor Lack of awareness in Seville pilot and a strong positive association of this factor in Thessaloniki. Positive association of Discouragement was found in Bilbao and MPH while a negative one was observed in Sussex. Finally, negative association of Other barriers was reported in Thessaloniki and Sussex.

4.3.2. Results from Post-Pilot Questionnaires

At the end of the intervention, a second assessment was performed to examine if the rankings provided to Persuasive Principles were still dependent on the variability of the socio-economic factors under study. Given the attrition rate on the POST survey we did not get any significant result on the Pearson's chi-squared test assuming the very restrictive Bonferroni correction. Removing this

correction threshold, Table 8 reports the significant *p*-values found and the associated power for this particular analysis. In this case, Similarity (P12), Conditioning (P3), Social Recognition (P14) and Reciprocity (P9) were found to be dependent on three socio-economic variables. As can be observed, City appeared in both PRE and POST analyses as a dependent factor for certain Persuasive Principles. Specifically, we found a double repetition of this factor affecting Similarity (P12) principle of persuasion in both snapshots. Besides, Confidence in Technology appears to be the variable with more PP associations. In fact, it is related to Conditioning (P3), which was one of the Persuasion Principles used in the treatments delivered. This result was not observed in the PRE analysis (see Table 6), yet it is related to a previous finding of the authors of this article in a previous research[33]. That is, interventions that try to affect the behaviour on energy efficiency through ICT-based equipment has an impact on the Confidence in Technology of the people subjected to this kind of interventions.

**Table 8.** Significant *p*-values (and the associated power) that ascertain the dependencies between Factors and Persuasive Principles for the POST survey.

| Persuasion Principles | City | Organisation Strategy | Confidence |
|---|---|---|---|
| **P3** | - | - | $p < 0.001$ $\beta = 0.6829225$ |
| **P9** | - | - | $p < 0.01$ $\beta = 0.5013527$ |
| **P12** | $p < 0.02$ $\beta = 0.9154948$ | - | $p < 0.05$ $\beta = 0.6605258$ |
| **P14** | - | $p < 0.05$ $\beta = 0.6637232$ | - |

As was done in the PRE-pilot, we have also investigated if there were dependencies among the different pilots (i.e., the cities where these are) and behavioural factors.

Pre-Contemplation, Contemplation and Action

As we observed in PRE, we continued finding differences among cities in these factors. A Kruskal–Wallis revealed a significant effect of City on Pre-Contemplation ($\chi^2(1) = 141.81, p < 0.01$), Contemplation ($\chi^2(1) = 58.559, p < 0.01$) and Action ($\chi^2(1) = 46.227, p < 0.01$). However, when we provided a post hoc test using Dunn's tests with Bonferroni correction we only saw the following significant differences reported in Table 9.

**Table 9.** POST-pilot: Post hoc analysis to identify where the differences are in the effect that we observed after running Kruskal Wallis test. In Pre-contemplation and Contemplation it seems that Bilbao scored differently than Greek pilots. Specifically, people in Greece scored higher on Pre-contemplation stage than the Spanish pilots. Interestingly, DEUSTO, SEVILLE and ALLIA reported the lowest scores in this stage of no-awareness. Regarding the Contemplation stage, there is a difference between the Spanish cities where DEUSTO again reported the highest average score in this stage of change and SEVILLE the lowest along with ALLIA and CERTH.

| | Pre-Contemplation | Contemplation | Action |
|---|---|---|---|
| **City** | Thessaloniki-Bilbao Pilea-Bilbao ($p < 0.008; r = 0.254$) | Seville - Bilbao Thessaloniki-Bilbao Cambridge -Bilbao ($p < 0.01; r = 0.261$) | - |

Intentions, Barriers and Confidence in Technology

A contingency analysis for these factors using G test, again because of the sample size, showed that all of them depend on the city significantly (whereas in PRE, Intentions did not depend on City).

Intentions ($\chi^2(12) = 23.735$, $p < 0.05$), Confidence ($\chi^2(12) = 38.48$, $p < 0.005$) and Barriers ($\chi^2(18) = 40.35$, $p < 0.002$). Figures 8–10 show the different distribution of responses depending of the city of origin that help us identifying which of them responded differently to the rest.

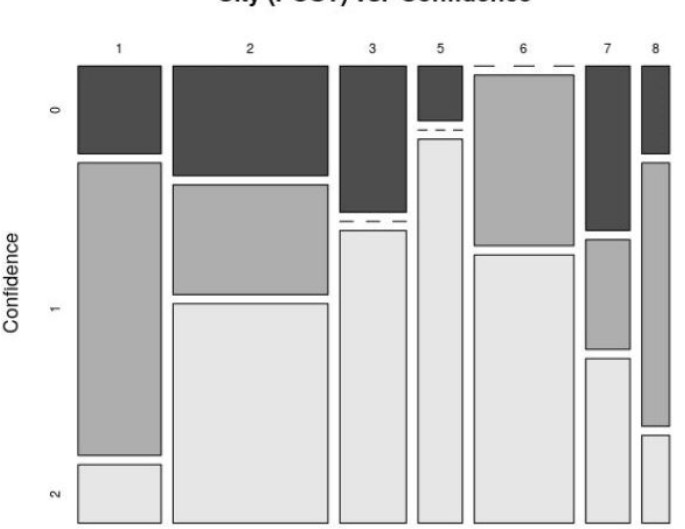

**Figure 8.** POST-pilot: Comparison of the distribution of confidence in technology among different pilots. Y axis: neutral (0), No Confidence (1) and Confidence (2).

According to the Pearson standardised residuals measures performed over the contingency data in Figure 8, we observed a negative association of Neutrality with Thessaloniki's employees. A strong positive association of No-confidence in technology reported in Seville pilot and a slight negative association in WEIZ building. Finally, we observed a strong negative association of the factor Confidence in technology in Seville while in MPH employees provided a positive association of this factor (i.e., MPH reported the highest rate of confidence in technology in a significant way).

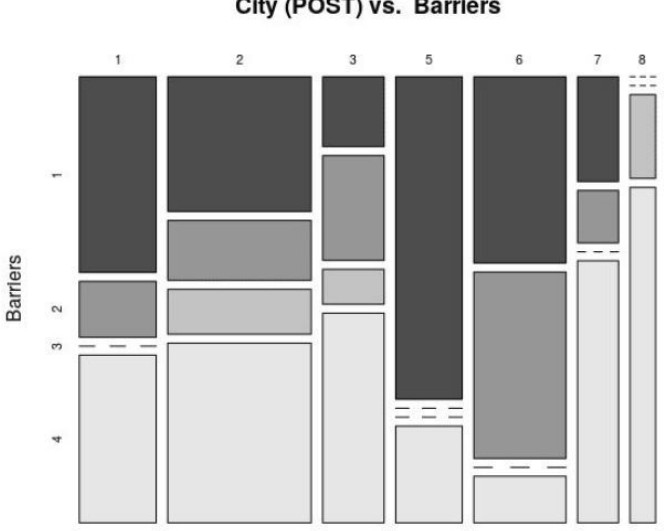

**Figure 9.** POST-pilot: Comparison of the distribution of barriers in each pilot. Y axis: Absentmindedness (1), Lack of Awareness (2), Discouragement from Peers (3) and Another Reason (4).

Pearson standardised residuals measures performed over the contingency data in Figure 9 provided a quite strong positive association of factor Absentmindedness in MPH and a strong positive association of factor Lack of awareness in Thessaloniki's employees. In this latter pilot, we also observed a negative association with Other barriers, meaning that tenants from this building identified clearly the barriers that hinder sustainable actions.

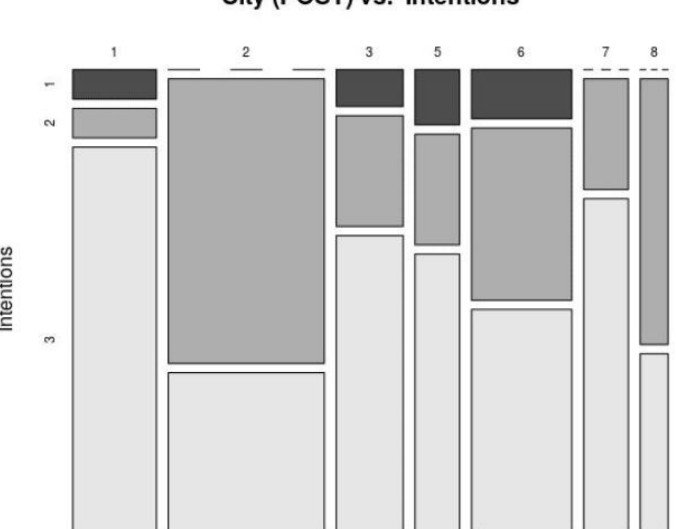

**Figure 10.** POST-pilot: Comparison of the distribution of intentions in each pilot. Y axis: Pre-Contemplation (1), Contemplation (2) and Action (3).

To conclude this analysis, Pearson standardised residuals measures performed over the contingency data in Figure 10 reported a positive association of Contemplation and a slight negative association of Action in Bilbao's building (DEUSTO). In Seville, however, we found a positive association of Action which means that the majority employees in this building reported to do actions in favour to the environment.

*4.4. Summary of Results*

Both the PRE and POST pilot surveys have been analysed in a descriptive and prescriptive manner. By comparing the results of the two surveys, the overall characteristics of the population remained more or less the same as has been reported in Section 4.2. It was really an interesting finding to observe that the results from the POST questionnaire pointed out to the same top three strategies that were best ranked in PRE: (Cause and Effect (P2), Conditioning (P3) and Self-monitoring (P11)). Indeed, these three strategies were the ones selected after the first intervention to inform the design of the different treatments delivered during the GreenSoul study.

Regarding the socio-economic factors that affect self-perceived usefulness of Persuasive Principles, "City", "Education" and "Initiative to join environmental campaigns" were identified as dependent factors over some Persuasion Principles in Pre-Pilot questionnaire. In the POST questionnaire, only City remained as an affecting factor to some PPs. Besides, we found that Confidence in technology and Organisational factors at workplace appeared to provide an effect in POST that were not observed in PRE. The Similarity principle ("People are more readily persuaded through systems that remind them of themselves in some meaningful way.") was the only PP observed in the results of both questionnaires always affected by City.

In terms of intentions to act in favour of the environment, it was relevant to find that general employees' pro-environmental intentions have shifted towards more active participation (Action)

than before the pilot intervention by 8.5%. Statistically speaking, we only found a difference in the Pre-contemplation stage between the responses of PRE and POST pilots questionnaires when treating the dataset as a whole (pooled samples). Thus, in general, at the end of the ICT-based intervention employees seemed to be more aware of the potential actions to do towards energy efficiency.

Furthermore, we performed an analysis of all the behavioural constructs break down by pilot (i.e., comparing previous and post responses to the questionnaire within different cities). On the one hand, such analysis helped improve our understanding of where the reported statistical difference in Pre-Contemplation was more accentuated. On the other hand, we were able to observe other differences in behavioural constructs that were not noticeable when doing pooled analysis. In DEUSTO (Bilbao, Spain), we found an strong positive difference in the Pre-contemplation stage between the responses provided in PRE and POST. Furthermore, employees in this pilot increased their scores in Contemplation and Action stages. This means that the overall effect of the ICT-based intervention was effective towards the purpose of making people to be more aware about the environment and contributing to mitigate energy inefficiency in an active manner. In SEVILLE (Spain), people increased their overall intentions in favour of the environment and it was the pilot with the higher number of employees responding to be in Action stage. WEIZ (Austria) and ALLIA (Cambridge, UK) also changed their Pre-contemplation stage towards higher stages of consciousness. In MPH (Pilea-Hortiatis, Greece), we did not find conclusive information on employees' intentions to behave in a more energy efficient manner. In the results, we do not provide information about ECOLUTION (Sussex, UK) nor CERTH (Thessaloniki, Greece) because of we did not find any differences on their results reported in PRE and POST time-frames. This result is relevant as these pilot sites were control conditions. Therefore, the potential effect of GreenSoul's intervention described in Section 2.1 is reinforced as no differences between PRE and POST were found on those pilots-buildings.

Barriers and Confidence in technology have been thoroughly studied in the analysis provided. At the end of the intervention, the majority of the interviewees reported to have less barriers to act pro-environmentally than in PRE stage. Besides, having a look at the contingency tables in PRE and POST we found that MPH moved from discouragement to absentmindedness as a primary factor hindering the adoption of energy efficient actions. In this Greek building, none of the respondents reported that one of the obstacles to behave energy efficiently were due to lack of knowledge nor people were found to be discouraged about their work-peers. This finding suggests that the intervention might have increased the energy-awareness significantly and reduced the discouragement about peers. If the main barrier, overall, that users encounter after the end of the intervention in MPH was occasional absentmindedness and not other, our hypothesis is that the GS intervention should have provided a positive effect. In the analysis we also observed how CERTH (control group) provided a strong association with the factor Lack of awareness in PRE and POST. This finding is indicative that these barriers steadily prevent employees to behave pro-environmentally at the workplace if they are not provided with any green-intervention.

Finally, we reported the impact of the Confidence in technology in the success or failure of ICT-based persuasive interventions. In POST, we found that "trust" seems to be an affecting factor for three PPs: Conditioning, Reciprocity and Similarity. As this finding was not observed in PRE questionnaire, we hypothesise that the GS intervention mediated this effect. However, this is only a exploratory result that deserves further investigation. In the same vein, it was also interesting to observe that MPH was the Pilot that increased its confidence in technology by far with regards to other pilots after the intervention. Conversely, SEVILLE's respondents always presented less confidence in technology at the beginning and end of the piloting regardless of the ICT-based intervention.

## 5. Implications on ICT-Based Interventions for Sustainability

The study has provided relevant results in terms of the design of future interventions based on ICT towards energy-efficiency in tertiary buildings. On the one hand, the results pointed out that Cause and Effect (P2), Conditioning (P3) and Self-monitoring (P11) were the top self-ranked Persuasive

Principles across all pilot buildings in UK, Spain, Greece and Austria. Observing that even after the intervention the employees kept choosing them as the top-strategies, that suggested that we were right selecting and delivering them in the different ICT-based experimental conditions. In the body of the literature, other researchers have reported that other principles of persuasion were equally effective for different user-profiles (e.g., praise [34], commitment [35], social approval [36] or normative feedback [37]). This confirmed that different socio-economic factors affect the self-reported principles of persuasion. Therefore, it is challenging to provide gold-standards without performing ex-ante user-research studies of the population that will receive them [38].

According to the results reported in this manuscript, we argue that the success or failure of an ICT-based strategy based on Similarity principle will depend on the city where it will be delivered. This finding is controversial as "City" is just a physical location. Therefore, we deem that behind the City factor, there are many hidden cultural factors that have not been captured in this questionnaire but that are affecting principles of persuasion (in this case, Similarity). In any case, the result appears to be right as sharing similar attitudes and traits or sharing membership in a group was found to be positively associated with liking the persuader and endorsing the persuasive message [39]. In fact, a study found that demographic and behavioural similarity between the source and recipients resulted in more positive behavioural changes [40].

The discouragement about peers as a barrier for doing pro-environmental actions was reduced after the intervention. This finding helps to demystify the idea that external factors play one of the key roles on demotivating people at workplace [9]. Besides, some pilots have completely changed the barriers encountered to behave energy-efficiently, which did not occur in control groups (e.g., CERTH). Interestingly in the same country, all MPH's employees reported a full understanding of the actions to do in favour of the environment. However, they regularly present absentmindedness. Therefore, it seems that frequent subtle feedback should be provided to employees and tenants to reduce forgetting once they are aware of the problem.

Finally, the results suggest that it is important to know the level of confidence in technology if an ICT-based intervention to change the people's behaviour was based on Conditioning, Reciprocity and Similarity principles. Thus, their success or failure might partially depend on the level of confidence in the technology of the end-users. Faith or trust on technology is usually well perceived by users almost without objection [41]. However, it is interesting to note some technological paradoxes in this context. Technology has a leading role as a solution but sometimes it is also part of the problem (e.g., technological artefacts conceived towards energy efficiency that do not really compensate during their lives the greenhouse gases emitted to produce them). Moreover, enhancements on energy efficiency can provoke increased demand for energy services or even misuse of them (Jevons paradox [42]). In the same vein, over-reliance on cutting-edge technology may bring undesired effects to pro-environmental behaviour and reduce the personal responsibility for action [43]. Conversely, we did not found these reported effects from the body of literature. In our study the pilots with higher level of confidence in technology at the end of the intervention (specially MPH) were found to be the ones with less barriers to behave energy efficiently or which intentions to do actions in favour of the environment were enhanced.

*Limitations and Mitigation Actions*

In this study we have provided useful insights about how user profiles can be used to inform more efficient ICT-based campaigns to promote sustainable practices in tertiary buildings. However, the work has some limitations that should be taken into consideration for generalisation purposes. First, the reduction observed from the PRE to the POST-pilot responses (one third) was high. However, it reflects typical attrition rates in similar research. In similar studies that involve ICT solutions, their usage is mostly at the discretion of the participant, and the participant has the option to discontinue its usage very easily. In any longitudinal study, where the intervention is neither mandatory nor critical to the participants daily activities (well-being, pro-environmental sustainability or energy efficiency), trial

participants will be lost well below of the 60–70% [44]. Although these rates were expected at the beginning of the study, we can not overlook that this represents the main limitation of the results. Beside the sample size reduction, the geographical distribution of the sample between PRE and POST (see Table 4) can be considered as another limitation. This latter issue might an effect on the reliability of the results presented; overall, the conclusions raised with pooled data (i.e., differences in Pre-Contemplation between PRE and POST). As a mitigation argument, we have declared and demonstrated with data that the overall sample was kept balanced in most socio-economic aspects (age, sex, education, etc.) between the two time-frames. More importantly, we provided an in-depth study within each pilot site in PRE and POST to draw specific conclusions for each pilot which reinforce the validity of the study. We acknowledge that with the data provided, it is not very easy to extrapolate results to the whole pan-European context. However, we argue that our results can shed light or give hints to designers, engineers, managers or other relevant stakeholders of tertiary buildings to decide which kind of interventions based on ICT and persuasion could be delivered according to their specific context and socio-economic profile. For that, in Appendix A, we provide a description of each building and the employees that work within.

Besides the previous shortcomings, hereafter we provide other limitations of the study which are typical on interventions that aim at forming new behaviours in the field. Lasting effect was not sufficiently addressed in the presented work. We have reported that subjects seem to have modified their behaviours at the end of the study. However, we were not able to measure whether the changes remained after the second snapshot. Future studies should address this limitation by providing washout periods at the beginning and at the end of the interventions. The selection of the population also presented difficulties to generalise results. The subjects participating in this study were selected using a RCT approach, and they were restricted to the participating entities in the European project. This approach could result in a bias for the generalisation of results since they might not be comparable to other cases in Western societies. Nevertheless, we found interesting results in public buildings where civil servants work (e.g., city councils). Therefore, we argue that the results may have an impact on this specific tertiary buildings in which schedules are fixed and there are groups of employees split in several offices were visitors get in and out daily. Finally, we report on the non-controlled effects that may hinder the internal validity of the study. It is difficult to be validated as we were investigating in a non-controlled environment. In-the-wild research related to sustainability usually entails several hidden and uncontrollable factors (e.g., global climate campaigns, pro-environmental media or news on climate emergency) which are out of the control of the researchers and may have had an impact on the responses the employees gave to the questionnaires. In essence, it is usual that researchers working on field studies report similar limitations as we have brought about in this section. It is important to emphasise the limitations, but also the validity and value that the empirical evidence provided in this manuscript have to the research community working on sustainability and pro-environmental behaviour change.

## 6. Conclusions and Outlook

To deliver a socio-economic tool that can be used to assess persuasion interventions in tertiary buildings, a two-step survey was designed and conducted in multiple buildings across Europe. Between the two steps, different persuasion strategies have been deployed in different premises through the GreenSoul project, which allowed the evaluation and validation of the survey as an assessment instrument. The results shed light on the importance of understanding user profiles both in socio-economic and behavioural terms to inform ICT-based campaigns to promote sustainable practices among employees. Beyond monetary incentives, which usually work in households, other engaging mechanisms need to be considered at tertiary buildings where employees are not aware of the impact of their everyday actions. Thus, recognition, certification, similarity or other carefully designed nudges to promote behavioural change. This work has found that a representative sample from the pan-European landscape agrees on selecting Self-monitoring, Cause and effect and Conditioning as the most promising

principles to engage people into energy-efficiency. As there are already quite a few studies exploring which persuasion approaches seem to work best on different contexts, our future steps will seek to understand if people prefer a variety of incentivisation mechanisms or if they stick with just a few of them. Besides, we will emphasise on understanding how the confidence in technology, as a mechanism to solve all environmental issues, impact on the type of engaging mechanisms to deliver.

**Author Contributions:** Conceptualisation, D.C.-M., A.C.T. and C.E.B.; methodology, D.C.-M. and O.K.-E.; validation and formal analysis, D.C.-M. and A.C.T.; writing—original draft preparation, D.C.-M. and A.C.T.; writing—review and editing, S.K., D.L.-d.-I. and J.M.A.; supervision, D.T. and D.L.-d.-I.; project administration, J.M.A. All authors have read and agree to the published version of the manuscript.

**Funding:** This work has been partially supported by the European Commission through the project HORIZON 2020-RESEARCH & INNOVATION ACTIONS (RIA)-696129-GREENSOUL. We also acknowledge the support of the Spanish government for SentientThings under Grant No. TIN2017-90042-R.

**Acknowledgments:** The authors would like to thanks all those that helped in the pilot deployments and follow ups. Special mention to Arazt Manterola and Mikel Solabarrieta.

**Conflicts of Interest:** The authors declare no conflict of interest.

## Abbreviations

The following abbreviations are used in this manuscript:

| | |
|---|---|
| ICT | Information and Commu Technologies |
| GS | GreenSoul |
| PP | Persuasive Principle |
| PS | Persuasive Strategy |
| TTM | Transtheoretical Model |
| PRE | Pre-Pilot |
| POST | Post-Pilot |
| RCT | Randomised Controlled Trial |

## Appendix A. Offices and Pilots Description

Throughout the whole manuscript we provide relevant outcomes from official GreenSoul pilot buildings. Therefore, this section shows a descriptive summary of the pilot sites aside with relevant data that help to extrapolate results to similar settings (a thorough description of these buildings and characteristics from employees and tenants can be found in [45]).

**WEIZ (Austria):** The pilot was deployed at the Energy and Innovation Centre of Weiz (W.E.I.Z.), which is an innovative and trendsetting business centre. Here, 34 entrepreneurs and organisations in the field of F&E, Economy and Education find attractive office and storage rooms, which are conceived after the latest cognitions and get professional support from the management of W.E.I.Z. The W.E.I.Z. is located in the centre of Weiz (Austria). The optimal infrastructure is supplemented by flexible room sizes and a sophisticated use and energy concept. The building campus is the largest Styrian impulse centre outside the capital city Graz. The opening hours for visitors are from 07:00 a.m. to 07:00 p.m. on weekdays and 08:00 a.m. to 12:00 a.m. on Saturdays (in case of a specific event opening hours are extended to 11:00 p.m.). The tenants have keys and can access at any time thus their occupancy can only be estimated.

**ALLIA (UK):** The Future Business Centre is owned and operated by Allia Ltd. It opened in November 2013. The Future Business Centre is a business innovation centre with a difference—to grow businesses that do good for society and the environment. More than just a set of workspaces for rent, it is a place where people can grow their ideas to make a difference in the world. It offers affordable, high quality workspace on flexible terms with specialist business support and an ethos of collaboration and innovation. The building has shared toilets, changing facilities and tea points. A reheat kitchen and eating area are located on the ground floor. There are currently 500 live "access cards" but these include contractors "hot deskers" and a sizeable "part-time staff".

**ECOLUTION (UK):** Affinity Sutton is one of the largest providers of affordable housing in England managing over 58,000 homes and properties in over 120 local authorities. They are a non-profit organisation for social purpose with commitments to reduce carbon emissions and increase energy efficiency across its own buildings and housing stock including tackling fuel poverty of residents. The pilot is settled in an open space area on the third floor of the Upton House, a three-storey-building, comprising offices and meeting rooms. Specifically, the pilot area is an non automated open space with several groups of workstations distributed across the floor. It is estimated that employees spend ~6 h in the office, 1 h in meetings and 1 h at lunch. Usually there are approximately 10 visitors per day.

**DEUSTO (Spain):** The pilot was held at ESIDE building, which was built in 1921 and houses the famous Faculty of Economics and Business Administration. Its neoclassical façade is 107 m in length, and it consists of a basement, ground floor and two floors. In 1996, the modern building was attached to its back. ESIDE hosts the Faculty of Engineering, DeustoTech (a research centre which belongs to Deusto Foundation) and DeustoKabi (a start-up incubator). Moreover, the new adjacent facilities for the Sports Degree have been opened in 2014. All these buildings share the electricity, water and heating systems. The people in the pilot range from researchers, technicians, project managers, accountants, working an average of 8 h per day. The daily operation hours of the building are weekdays from 08:00 a.m. to 08:00 p.m. and on the weekends from 09:00 a.m. to 02:00 p.m. The employees can access the building at this time. The daily occupancy is estimated on 120 employees and 10 visitors per day.

**SEVILLE (Spain):** The pilot was held at the Institute of Statistics and Cartography of Andalusia, which was built in 1992 for the World Exposition that was hosted in Seville. The building was the New Zealand Pavilion that was built as a touristic building. After the end of the World Exposition, the pavilion was acquired by the public regional government and they built offices inside to host the whole Institute of Statistics and Cartography of Andalusia. The building consists of basement, three floors and roof. The pilot has an area of 3.529 m$^2$. One of the main challenges was to rebuild the building into offices and the major project was related to the adaptation of the air conditioning to current usage conditions and energy requirements. Due to the fact that the existing facility was designed for an unlimited period of time and characteristics of use, it was necessary to design a completely new air conditioning system. The offices are used by the public administrators and civil servants. The vast majority of this personnel spend at least 7 h in the office.

**MPH (Greece):** The pilot was held at the Pilea-Hortiatis municipality hall buildings. The Municipality of Pilea-Hortiatis is based in a quite new building operating from 2010 on. A variety of retrofitting actions have been performed, updating the buildings to energy category B according to KENAK (Greek Regulation for the Energy Efficiency of Building), reducing their operational energy consumption. Furthermore, the roof of the Municipality Hall buildings is partially covered with photovoltaic (PVs) panels and the energy produced is sold to the National Electricity Provider Company (Public Power Corporation S.A). The Municipal Hall is open to the general public 5 days a week from 7:00 a.m. to 16:00 p.m., while Building B is open in the afternoons till 21:00 a.m. as it hosts the music school and the concert and conference halls. The offices are used by the administrative personnel and members of the Municipal Council or civil servants.

**CERTH (Greece):** The pilot was held at the Information Technologies Institute (ITI) central building at the Centre for Research Technology Hellas in Thessaloniki. The building was constructed in 2000, and consists of ground floor, two floors (one of which was constructed later as an extension with a metal foundation) and two undeground parking levels. Average working hours are from 09:00 to 17:00 and hosts mainly ICT-related activities, including also administrative ones. The building has also server rooms, dedicated labs, and meeting/conference rooms that are used based on daily needs. Even though there is a BMS available, all building assets (HVAC, Lights, Appliances) are fully controlled by the end-users.

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
