# Peer review of "Socio-Economic Effect on ICT-Based Persuasive Interventions Towards Energy Efficiency in Tertiary Buildings"

_energies, doi:10.3390/en13071700_

Round 1

Reviewer 1 Report

This is an interesting and well written paper. Some typos to correct, e.g., "even thought 15 principles were originally". The methodology and design of the survey questions are well thought out. Two comments: (1) 350 people were surveyed who are across 6 cities and 4 UE countries. Is the population/cities representative of EU as a whole? How were they chosen? based on project participating? (2) The pre and post survey were a snapshot. Would the effective persuasion strategies continue to be effective along time?

The link to PRE survey has an extra "/" -  https://doi.org/10.5281/zenodo.2610102/

Reviewer 2 Report

Very interesting study; my main concern has to do with the construction of the 2 samples, see comments below.

Title: consistency on the first letter of all words included in the title (either capital or small). 

P1L5: "...and the impact that it has to build..." not clear what you mean.

P1L12-14: "...that shed light on understanding which of them..." not clear where you refer to; "that shed" refers to principles of persuation? Perhaps repharse sentence, to be more clear. 

General comment: you change between past and present tense; be consistent on the tense used. 

P1L16: "...a recommendation-based intervention through ICT were delivered..."; either remove "a" and change "intervention" to "interventions" OR change "were" to "was".

P1L16-17: "The findings from the PRE-pilot stage were used to refine the POST-pilot survey."; what are these pilot stages/surveys? Something different from the 2 main surveys? Not really clear here.

P2L32: delete "as"

P2L33: change to "beliefs"

P2L44: "...with the most well known..." add "being"

P2L66: Define acronym "ICT" in its first appearance

P3L82: "7", "six", "4"; be consistent with the use of numbers

P3L114-115: You mean "...and a decision support system..."? The connection of the sentence before and after the comma is not clear.

P4L143: "Procedure an evaluation"; you mean "..of an..."?

P5L167 delete "also", you already use "Additionally"

P6: Table, "Cooperation & Liking": What does "(Coke)" refer to? Is it presented as an example? Not clear.

P6: Table, "Social proof": delete 1st word ("Is")

P7 v8: "You receive information about the people behind energy-related data collection". Just to be clear: you receive information concerning the people that collect energy-related data? Is this the case? 

P7 v13: replace "...have..." with "has", since you refer to "feasibility"

P7L188: "...of the persuasive..."

P8L190: replace "provided" with "provide"

P8L197: define what acronym "CERTH" stands for

P8L200: Not clear what you mean with "...hence allowing the existence of control groups..."

P8 - Survey setup: here I detect the major deficiency of this work so far.

You have an initial sample of 303 people, showing great differences between the participating countries: from 118 participants from the UK to 18 from the Austria. In addition, the POST survey consists of a much smaller number, with differences not only between the countries (e.g. 43 from Spain, 13 from the UK), but also in relation to the PRE study: e.g. UK had 118 participants in the PRE study and only 13 in the POST study. And, as it can be seen in the Results section, the results are based on pooled data, while the PRE and POST results are compared, although the synthesis of the sample (at least country-wise) is totally different between them. Thus, do you believe that in terms of survey methodology it is correct to pool this data, when each country is not represented evenly/ proportionally? Is it correct to compare the PRE and POST samples, with so large differences in the sythesis of their sample? (refering to section 4.2 Descriptive Analysis) 

P8 - Survey setup: Why where the specific buildings from the specific countries selected as samples? Only because they were the ones participating in the same European project? Are there any similarities/ differences between them, making you select buildings from the specific countries? 

P8 - Table 4: Why is the decrease in the responce rate so large specifically for the UK sites? Following my above comment, the 1st sample consist of more than 1/3 British, while in the 2nd sample the UK represents around 12%; doesn't this create a problem in directly comparing the PRE and POST samples? 

P10 Figure 3: Figure not of good quality

P11 Figure 5: Not clear what each box represents; perhaps the numbering at the side of the boxes (i.e., 1, 2, 3) should be larger. 

P11L299: "Overall because of the large effect size of this statistical finding"; what does this sentence refer to? Not clear, perhaps rephrase. 

P13L359: "it is provided a Post-hoc analysis" change to "a Post-hoc analysis is provided"

P16L417: "...for Pilea"

P18L431: replace "form" with "from"

P18L440: replace "Similarity..." with "Similarly..."

P19L476: "Our" change to "our"

P19L498-499 "...list top list..."; make correction

P19L503: replace "it" with "its"

P19L505: correct "...a the..."

P19L509: replace "an" with "a"

P20L532: replace "found" with "find"

P20L545: "This work has found that a representative sample..."; based on the above comments, can you present this sample as "representative"?

P20L532: Please include all abbreviations presented in the document; e.g what does CERTH-ITI stand for?

Please include a limitations section, presenting the limitations of the study; perhaps the issues concerning the samples structure (as described in previous commnets) could be addressed here).   

Round 2

Reviewer 2 Report

My comments have been properly addressed.